# Assessment of minority frequency pretreatment HIV drug-resistant variants in pregnant women and associations with virologic non-suppression at term

Ceejay L. Boyce[1,2], Ingrid A. Beck[1], Sheila M. Styrchak[1], Samantha R. Hardy[1], Jackson J. Wallner[1], Ross S. Milne[1], R. Leavitt Morrison[3], David E. Shapiro[3], Esaú C. João[4], Mark H. Mirochnick[5], Lisa M. Frenkel [1,2,6]*

1 Center for Global Infectious Disease Research, Seattle Children's Research Institute, Seattle, Washington, United States of America, 2 Department of Global Health, University of Washington, Seattle, Washington, United States of America, 3 Center for Biostatistics in AIDS Research, Harvard T.H. Chan School of Public Health, Boston, Massachusetts, United States of America, 4 Infectious Diseases Department, Hospital Federal dos Servidores do Estado, Rio de Janeiro, Brazil, 5 Boston University School of Medicine, Boston, Massachusetts, United States of America, 6 Departments of Pediatrics, Laboratory Medicine and Pathology and Medicine, University of Washington, Seattle, Washington, United States of America

* lfrenkel@uw.edu

## Abstract

### Objective

To assess in ART-naïve pregnant women randomized to efavirenz- versus raltegravir-based ART (**IMPAACT P1081**) whether pretreatment drug resistance (**PDR**) with minority frequency variants (<20% of individual's viral quasispecies) affects antiretroviral treatment (**ART**)-suppression at term.

### Design

A case-control study design compared PDR minority variants in cases with virologic non-suppression (plasma HIV RNA >200 copies/mL) at delivery to randomly selected ART-suppressed controls.

### Methods

HIV *pol* genotypes were derived from pretreatment plasma specimens by Illumina sequencing. Resistance mutations were assessed using the HIV Stanford Database, and the proportion of cases versus controls with PDR to their ART regimens was compared.

### Results

PDR was observed in 7 participants (11.3%; 95% CI 4.7, 21.9) and did not differ between 21 cases and 41 controls (4.8% vs 14.6%, p = 0.4061). PDR detected only as minority variants was less common (3.2%; 95% CI 0.2, 11.7) and also did not differ between groups (0% vs. 4.9%; p = 0.5447). Cases' median plasma HIV RNA at delivery was 347c/mL, with most (n =

**Data Availability Statement:** Nucleotide consensus sequences are available in the NCBI Genbank under accession numbers MZ475744-

MZ475806. Illumina data are available in the NCBI Sequence Read Archive under BioProject number PRJNA743320.

**Funding:** Support for this analysis and overall support for the International Maternal Pediatric Adolescent AIDS Clinical Trials Network (IMPAACT) was provided by the National Institute of Allergy and Infectious Diseases (NIAID) with co-funding from the Eunice Kennedy Shriver National Institute of Child Health and Human Development (NICHD) and the National Institute of Mental Health (NIMH), all components of the National Institutes of Health (NIH), under Award Numbers UM1AI068632 (IMPAACT LOC), UM1AI068616 (IMPAACT SDMC) and UM1AI106716 (IMPAACT LC), and by NICHD contract number HHSN275201800001I. Additional support came from the Molecular Profiling and Computational Biology Core of the University of Washington Fred Hutch Center for AIDS Research [P30 AI027757]. C.B. received support from the NIH under Grant Award Number T32 AI007509. The content is solely the responsibility of the authors and does not necessarily represent the official views of the NIH. The funders had no role in study design, data collection and analysis, decision to publish, or preparation of the manuscript.

**Competing interests:** The authors have declared that no competing interests exist.

19/22) showing progressive diminution of viral load but not ≤200c/mL. Among cases with viral rebound (n = 3/22), none had PDR detected. Virologic non-suppression at term was associated with higher plasma HIV RNA at study entry (p<0.0001), a shorter duration of ART prior to delivery (p<0.0001), and randomization to efavirenz- (versus raltegravir-) based ART (p = 0.0085).

## Conclusions

We observed a moderate frequency of PDR that did not significantly contribute to virologic non-suppression at term. Rather, higher pretreatment plasma HIV RNA, randomization to efavirenz-based ART, and shorter duration of ART were associated with non-suppression. These findings support early prenatal care engagement of pregnant women and initiation of integrase inhibitor-based ART due to its association with more rapid suppression of plasma RNA levels. Furthermore, because minority variants appeared infrequent in ART-naïve pregnant women and inconsequential to ART-suppression, testing for minority variants may be unwarranted.

## Introduction

Concomitant with the increased utilization of antiretroviral therapy (**ART**), pretreatment drug resistance (**PDR**)—defined as resistance detected in individuals qualifying for the initiation or reinitiation of first-line ART—has increased globally [1]. PDR can compromise the effectiveness of ART regimens [2–6]. Perinatal HIV transmission, which has been found to peak in the final weeks of gestation and peripartum period [7], has been associated with PDR and maternal plasma HIV RNA load [8]. Therefore, to minimize perinatal transmission, HIV-infected pregnant women presenting for care late in pregnancy should be prescribed ART regimens that rapidly suppress viral replication.

As part of HIV standard-of-care in the United States and in other high-resource settings, Sanger sequencing (also known as genotyping, consensus or population sequencing) is performed to guide ART selection with avoidance of drugs to which the virus is resistant [9]. However, Sanger sequencing does not reliably detect HIV drug resistance mutations present at <20% of an individual's HIV quasispecies (i.e., "minority variants"). Dual-class PDR—including minority variants—has been associated with poorer rates of suppression of HIV replication by efavirenz (**EFV**)-based ART [10]. Thus, we hypothesized that minority variants would be associated with virologic non-suppression at term.

NICHD P1081 was a multicenter, randomized trial comparing raltegravir (**RAL**)- vs. EFV-based ART regimens in HIV-infected, treatment-naïve pregnant women initiating ART during pregnancy [11]. In the present study, we aimed to compare the prevalence of PDR minority variants in women with versus without virologic non-suppression (plasma HIV RNA >200 copies/mL at delivery) to assess if PDR mutations affected the efficacy of the P1081 ART regimens.

## Methods

ART-naïve pregnant women (n = 408) enrolled between 20–36 weeks gestation were randomized to RAL or EFV with lamivudine (**3TC**)/zidovudine (**ZDV**) or locally available alternative nucleoside reverse transcriptase inhibitors (**NRTI**s) in the P1081 trial. Written informed consent was obtained for the overall study including assessment of HIV drug resistance from all

participants at an initial screening visit in accordance with procedures approved by Human Subjects Committees. Clinical drug resistance testing was performed at screening or enrollment, but results were not required for assignment of study drugs. Once results were available, clinicians could modify any or all antiretrovirals based on the woman's HIV genotype. While women who had PDR at entry were excluded from the primary P1081 study analyses [11], all participants were followed to term for secondary analyses and were eligible for inclusion in this case-controlled study. "Cases" were women with plasma HIV RNA load >200 copies/mL at delivery who had been prescribed ART for ≥14 days. "Controls" were study participants with ART-suppression at delivery selected randomly from matched study sites—to avoid confounding by social and environmental factors specific to clinical sites—at a ratio of 1:2.

To study minority PDR variants, the region of HIV that encodes reverse transcriptase was sequenced retrospectively using the Illumina MiSeq platform. Briefly, RNA was extracted from 0.14-1mL plasma from study screening or entry using the QIAamp viral RNA mini kit (Qiagen, Hilden, Germany). Reverse transcription with SuperScript III First-Strand Synthesis System (Invitrogen, Carlsbad, California) used a primer consisting of a HIV-specific sequence (S1 Table) followed by an 8bp string of random nucleotides ("unique molecular identifier (**UMI**)") and a universal 24bp Illumina reverse adapter sequence. The cDNA was purified using beads (Agencourt Ampure XP, Beckman-Coulter, Beverley, Massachusetts) and amplified for 45-cycles with primers with indexing adapter sequences (S1 Table) using a high-fidelity PCR kit (FastStart High-Fidelity PCR system, Roche Diagnostics, Mannheim, Germany). Samples were indexed (IDT for Illumina Nextera DNA Unique Dual Indexes, Illumina, Inc., San Diego, California) then pooled together and sequenced bidirectionally on an Illumina MiSeq (MiSeq Reagent Kit v3, Illumina, Inc.).

Sequences were processed as previously described [12], with additional steps to generate consensus sequences using UMIs. Three or more high-quality sequences with identical UMIs were collapsed into a single consensus sequence and aligned to the HXB2 HIV reference sequence using the Burrows–Wheeler algorithm [13]. Consensus sequences were examined for nucleotide variants associated with resistance to NRTIs and NNRTIs as defined by Stanford's HIV Drug Resistance Database [14,15]. Participants with fewer than 100 consensus sequences, each from one viral template, were excluded to ensure adequate representation of minority variants within their viral quasispecies. Potential cross-contamination between participants' sequences and concordance of Illumina and Sanger sequencing data was assessed phylogenetically. The variant calling pipeline is available at https://github.com/MullinsLab/drm-snp-calling.

Reverse transcriptase mutations with a Stanford Database genotypic resistance score ≥15 to any antiretroviral drug included in each woman's study regimen were analyzed. In addition, minority (<20% of the viral quasispecies) and majority (≥20%) frequency variants containing mutations conferring resistance to NRTI and NNRTI with a score <15 were tallied and reported. The proportion of cases versus controls with PDR variants to the P1081 ART regimens were compared by Fisher's Exact test. Univariate analyses were also performed to compare age, CD4 cell count at study entry, plasma HIV RNA load at study entry, and time on ART until delivery between cases versus controls by the Mann-Whitney test. Statistcal significance was defined as a two-sided p-value <0.05 for all analyses. All statistical analyses were conducted using GraphPad Prism software version 9.3.1.

## Results

Among 408 women enrolled and prescribed ART, 26 women had virologic non-suppression at delivery, 4 were excluded due to <14 days on ART, and 22 "case" women were studied and

**Table 1. Comparison of Pre-ART HIV drug resistance mutations detected by Illumina sequencing in participants with virologic non-suppression at term vs. ART-suppressed control women.**

| Variable | Total (N = 69) | Cases (N = 22) | Controls (N = 47) | P value |
|---|---|---|---|---|
| **Distribution of participants by randomization arm** | | | | 0.0085[a] |
| RAL | 29 (42.0) | 4 (18.2) | 25 (53.2) | |
| EFV | 40 (58.0) | 18 (81.8) | 22 (46.8) | |
| **Age (years), median (range)** | 25 (14–44) | 27 (18–38) | 24 (14–44) | 0.3886[b] |
| **Gravidity, median (range)** | 3 (1–9) | 3 (1–7) | 3 (1–9) | 0.1384[b] |
| **CD4+ lymphocytes values (cells/uL), median (range)** | 337 (1–917) | 289 (34–865) | 369 (1–917) | 0.4561[b] |
| **Plasma HIV RNA at entry ($\log_{10}$ copies/mL), median (range)** | 4.34 (2.73–5.92) | 4.88 (3.39–5.92) | 4.05 (2.73–5.89) | <0.0001[b] |
| **Time on ART until delivery (days)** | 70 (19–156) | 43 (19–127) | 86 (24–156) | <0.0001[b] |
| **Plasma HIV RNA trend at delivery** | | | | |
| HIV RNA ART-suppressed | 47 | | 47 (100) | |
| HIV RNA progressively decreasing | 19 | 19 (86.4) | | |
| HIV RNA rebounding | 3 | 3 (13.6) | | |
| **Total N (%) with HIV drug resistance mutations with Stanford Score ≥15** | 7 (11.3) | 1 (4.8) | 6 (14.6) | 0.4061[a] |
| Majority variant(s) only (≥20% frequency) | 4 (6.5) | 0 | 4 (9.8) | |
| Minority variant(s) only (<20% frequency) | 2 (3.2) | 0 | 2 (4.9) | 0.5447[a] |
| Majority and minority variants | 1 (1.6) | 1 (4.8) | 0 | |
| Single mutation | 3 (4.8) | 0 | 3 (7.3) | |
| ≥2 mutations | 4 (6.5) | 1 (4.8) | 3 (7.3) | |
| EFV-resistance only | 2 (3.2) | 0 | 2 (4.9) | |
| ZDV-resistance only | 2 (3.2) | 0 | 2 (4.9) | |
| 3TC-resistance only | 0 | 0 | 0 | |
| NRTI- and NNRTI-resistance | 3 (4.8) | 1 (4.8) | 2 (4.9) | |
| **Additional N (%) with HIV drug resistance mutations with score <15** | 11 (17.7) | 4 (19.0) | 7 (17.1) | 1.0000[a] |

Abbreviations: ART, antiretroviral therapy; RAL, raltegravir; EFV, efavirenz; ZDV, zidovudine; 3TC, lamivudine; pMTCT, prevention of mother-to-child transmission; NRTI, nucleoside reverse transcriptase inhibitor; NNRTI, non-nucleoside reverse transcriptase inhibitor.

[a] Fisher's Exact test.

[b] Mann-Whitney test.

compared to 47 "control" women with ART-suppression. The 22 case women's entry plasma HIV RNA loads were higher (p<0.0001), they started ART closer to the time of delivery (p<0.0001), and were disproportionately randomized to EFV-based ART (p = 0.0085) compared to control women (Table 1). Among women with viremia at delivery the median plasma HIV RNA was 357 copies/mL (interquartile range: 250–752 copies/mL). While their HIV RNA load was >200 copies/mL—meeting the definition of virologic non-suppression of this study—at delivery, longitudinal plasma HIV RNA values of 19/22 (86.4%) case women revealed a decreasing trajectory from study entry. The other three case women experienced an increasing plasma HIV RNA or virologic "rebound" at term. Of the 69 study entry specimens examined in this study, HIV sequences were successfully amplified from 66 specimens (22 cases and 44 controls). Phylogenetic validation showed all Illumina sequences grouped with their matching Sanger sequences, and bioinformatic analyses indicated the median number of viral templates sequenced by Illumina was 375 (interquartile range: 229–574), with too few HIV templates (<100) to evaluate minority variants from one case (who was slow to suppress at term) and three controls; these four women were excluded from the analyses.

Overall, PDR was detected in 7/62 (11.3%) women by Illumina sequencing, but did not differ significantly in prevalence between the cases and controls (1/21; 4.8% vs. 6/41; 14.6%,

**Table 2. Pre-ART HIV drug resistance mutations with stanford score ≥15 detected by Illumina[a] sequencing and their frequencies in individuals' HIV quasispecies.**

| Participant | Case or Control | Study ART | Days on ART | HIV Drug Resistance Mutations with Stanford score ≥15 to ARV in woman's ART regimen | HIV Drug Resistance Mutations with Stanford score ≥15 to ARV not in woman's ART regimen |
|---|---|---|---|---|---|
| BR8135C_2017 | Case | EFV+3TC+ZDV | 30 | EFV: K101E 48%, V106A 2.2%, Y181C 58%<br>ZDV: T215S 58% | |
| BR8392K_2017 | Control | EFV+3TC+ZDV | 99 | ZDV: M41L 96.5% | |
| BR0310I_2016 | Control | RAL+3TC+ZDV | 45 | | K103N 90%, G190A 100% |
| BR8715D_2018 | Control | RAL+3TC+ZDV | 63 | 3TC: M184V 5.2%[b] | G190S 5.2% |
| BR8325L_2017 | Control | RAL+3TC+ZDV | 136 | ZDV: T215N 88.5%, T215D 10.9% | K103N 99.7%, P225H 99.4% |
| TZ7944L_2017 | Control | RAL+3TC+ZDV | 156 | ZDV: K70R 4.4% | |
| TZ7750K_2017 | Control | RAL+3TC+ZDV | 143 | | K103N 99.1% |

Abbreviations: ART, antiretroviral therapy; EFV, efavirenz; RAL, raltegravir; 3TC, lamivudine; ZDV, zidovudine.

[a] One control woman had M230I mutation by Sanger sequencing but no resistance mutations were detected by Illumina sequencing.

[b] BR8715D had ZDV-resistance mutation M41L detected by Sanger sequencing, which was not detected by Illumina sequencing.

p = 0.4061). Among these seven women, six had PDR detected at study enrollment by Sanger sequencing. The majority variant PDR genotypes were concordant for all but one case woman; she had NRTI PDR (M41L) by Sanger and only minority variants (M184V at 5.2% and G190S at 5.2%) detected by Illumina (Table 2). An additional control woman without resistance detected by Illumina had M230I by Sanger sequencing.

The case woman with PDR was randomized to the EFV arm and she did not demonstate viral rebound, but was slow to suppress. She had three mutations conferring resistance to EFV (K101E at a frequency of 48%, V106A at 2.2% and Y181C at 58%) and one mutation associated with ZDV resistance (T215S at 58%) (Table 2). Several additional case women (n = 4/21; 19%) had PDR mutations (either V106I and/or V179D) associated with resistance to NNRTI but their sequences were classified by Stanford's Database as "no" or "potential low-level HIV drug resistance to EFV" even when the two mutations were combined. These mutations were all detected at high frequencies (median 99.98%, range 69–100%), with three in the EFV arm and one in the RAL arm. The three women with viral rebound did not have PDR.

Among the control women, PDR was detected in 6/41 (14.6%). NNRTI mutations were detected in 4 (9.8%), two of these also had NRTI mutations (M184V (3TC) and T215N/D (ZDV)). NRTI mutations alone were detected in 2 (4.9%) women; M41L and K70R (ZDV) (Table 2). Among the six control women with PDR, mutations were majority frequency variants in 4 (median 99.4%, range 96.5–100%) and minority frequencies alone in 2 (M184V and G190S at 5.2%, and K70R at 4.4%). All with NNRTI DR were randomized to RAL-based ART. Among the four with NRTI resistance, two with ZDV resistance were prescribed ZDV+3TC+RAL, one with ZDV resistance was prescribed ZDV+3TC+EFV, and the one with 3TC resistance was prescribed ZDV+3TC+RAL (Table 2). Additional PDR in the controls included NNRTI mutations with Stanford HIV Drug Resistance Database scores of <15 to each study drug (K101H, V106I, V179D, H221Y) in nine women (not shown in table), including two—BR8392K_2017 and BR0310I_2016—with additional higher scoring mutations.

Few participants had PDR comprised of only minority variants that conferred resistance to their ART at term; no cases and two controls. These two women had NRTI minority variants

(K70R (4.4%) or M184V (5.2%) and G190S (5.2%)) and both were prescribed ZDV+3TC +RAL. No NNRTI minority variants were detected in women randomized to EFV. A comparison of the prevalence of minority variants between case vs. control women, performed to infer an effect on virologic outcome, did not detect a significant difference (0/21 vs. 2/41; p = 0.5447).

## Discussion

The primary findings in this study are that: (1) this population of HIV-infected, treatment-naïve pregnant women had a moderate level of PDR (11.3%), but too few (n = 2/62) with PDR minority variants alone (i.e., without concomitant majority frequency variants) capable of compromising their ART regimen to accurately assess their impact on ART-suppression; (2) virologic non-suppression at term was associated with higher pre-ART plasma HIV RNA load, shorter durations of ART prior to delivery, and randomization to EFV-based ART; and (3) study adherence appeared to be "high" with relatively few women experiencing virologic rebound at delivery.

Overall, we observed a moderate level of PDR (11.3%) to NNRTI or NRTI classes in this study population. This level of drug resistance is among the lower rates (10–30%) reported for women living with HIV [16], particularly among women of childbearing potential in sub-Saharan Africa [12]. However, the majority of study participants (41/62, 66.1%) were Brazilian, and other studies of Brazilians report similar PDR rates [16–19]. In this study, PDR majority and/or minority frequency variants were not associated with virologic non-suppression, most likely due the rarity of dual class resistance to the women's ART in this population, which has been associated with high rates of virologic non-suppression to EFV-based ART [10,20]. Moreover, very few study participants had NRTI PDR which theoretically could increase the risk of non-supprssion in women randomized to either an EFV- or RAL-based ART regimen, and only a few cases of virologic rebound occurred, suggesting that most non-suppression at delivery was due to inadequate duration of ART, especially EFV-based ART, to achieve plasma viral loads to <200c/mL, as was suggested by our analyses.

Among the case women, only one had PDR (Stanford score >15) to her ART regimen (EFV+ZDV+3TC) and she was slow to suppress. She had majority and minority frequency mutations to EFV and to ZDV. Her clinical genotyping results became available just prior to term without time to change her ART regimen. Dual class PDR, which across other studies has been associated with viral non-suppression and virologic failure [10,20], was infrequent in this cohort (total n = 3). Besides the one case woman, the other two women with dual class resistance were randomized to RAL-based ART and, as in others in this study with NNRTI resistance randomized to RAL, these women had HIV ART-suppression at term. This observation differs from one recent report where NNRTI mutations were associated with increased risk of virologic failure on integrase inhibitor-based treatment [21], but these study participants had failed EFV-based ART, and it is possible that some with NNRTI resistance continued patterns of non-adherence.

While the study enrolled women in their last trimester of pregnancy, most achieved ART suppression by delivery, likely due to good adherence to ART and the relatively rapid decrease in plasma HIV RNA associated with integrase inhibitor-based ART [22,23]. Among those with virologic non-suppression at term, most had longitudinal specimens showing progressively decreasing viral loads and had low-level viremia at delivery. These women had shorter duration of ART—due to later enrollment—and had higher plasma RNA loads at study entry. The persistence of viremia at delivery, mostly at low levels, was likely due to production of virus particles from the relatively larger proviral populations of women presenting with higher

plasma HIV RNA levels and not from full rounds of viral replication. By delivery the viremia observed in women with non-suppression was at levels associated with a low risk of mother to child transmission [24,25]. Our findings and data from the DolPHIN-2 and VESTED trials, which compared EFV-based ART to integrase inhibitor-based ART in pregnant women [26,27], support the use of integrase inhibitor-based ART in pregnant women to suppress HIV RNA levels more quickly thus reducing the risk of perinatal transmission.

NNRTI resistance alone and its effects on EFV-based ART were not adequately evaluated in this study as the one case women with Y181C also had T215S in her pretreatment genotype. Studies have observed that PDR consisting of the most prevalent NNRTI-associated mutation K103N [14] alone is not associated with virologic failure in people treated with EFV+tenofovir +3TC [10,20], which is noteworthy as it minimizes the role of PDR as a primary cause of virologic failure. These data deemphasize the need for public health officials to focus on PDR, but to instead address other contributors to virologic non-suppression or failure.

NRTI resistance alone was detected infrequently (2/62, 3.2%) in our study population which is concordant with NRTI PDR surveillance data reported by others [17,19]. Nearly all the women with NRTI resistance in this study population had been randomized to RAL suggesting that RAL-based ART was effective in suppressing ZDV- or 3TC-resistant viruses with majority or minority frequency PDR. These observations support models suggesting that testing for PDR prior to integrase-inhibitor ART is not cost-effective [28]. Importantly, in this study no participant had PDR to both ZDV or to tenofovir and 3TC. Thus, while studies suggest integrase inhibitor-based ART may effectively suppress replication of virus with resistance to both tenofovir and 3TC [29,30], elevated rates of virologic failure with dolutegravir monotherapy infer that sensitivity to at least one NRTI adds potency to treatment with integrase inhibitors [31–33]. The low prevalence of NRTI only PDR minority variants and the small size of our study limited our power to detect an effect of minority variants on virologic non-suppression.

Illumina sequencing, while detecting a greater number of mutations compared to Sanger sequencing, missed two mutations detected by Sanger in controls that showed majority peaks in the chromatograms and clustered with Illumina sequencing in phylogenetic analyses. This discrepancy was most likely due to primer bias or due to sequencing of few but unrepresentative sequences by Sanger. The two specimens with discordant genotypes had HIV RNA loads of 16,408 copies/mL and 249,542 copies/mL, and they had 154 and 1,362 HIV templates sequenced by Illumina, respectively. Thus, it is more probable that primer bias minimized detection of viral templates with these genotypes. Potential primer bias is an important limitation of this and perhaps other studies that use next-generation sequencing due to the need for multiple primers to amplify and sequence shorter regions of a hypervariable HIV genome.

In addition to potential primer bias, this study had several other limitations. Our power to assess the prognostic value of minority drug resistance variants on virologic non-suppression was limited by the small sample size of our study coupled with the rarity of minority drug resistance variants detected in this population. While NNRTI-associated resistance mutations were the most common variants detected, the small sample size and imbalanced randomization of participants with NNRTI PDR to RAL-based ART precluded analysis of the risk of NNRTI PDR—majoriy and/or minority variants—on EFV-based ART efficacy. Lastly, we did not evaluate the viral quasispecies at delivery which prevented analyses of the selection dynamics of pretreatment minority variant genotypes during ART. However, even if our study design had included genotyping at term, the paucity of minority PDR among case women would have limited our ability to draw any conclusions regarding PDR variant selection.

Our findings that higher HIV RNA loads at study enrollment, a shorter duration of ART, and randomization to EFV were associated with increased risk of non-suppression at delivery,

confirms the primary analysis of the P1081 study [11] and other studies [8,34,35]. The lower rate of ART-suppression associated with shorter durations of ART was seen primarily in those randomized to the EFV-arm. This observation underscores the importance of women engaging in prenatal care early in pregnancy and, when presenting to care late in pregnancy, the importance of initiating potent ART regimens that rapidly suppress HIV replication; reinforcing the guidelines' recommendation to administer integrase inhibitor-based ART in pregnancy [36].

## Conclusion

In summary, while HIV drug resistance was moderately common in pregnant women qualifying for first-line ART in this substudy, minority frequency PDR variants alone were rare. In this study, PDR composed of majority or minority variants was not associated with non-suppression possibly due to few instances of dual-class antiretroviral PDR and the imbalanced randomization of women with NNRTI PDR to RAL-based-ART. Given minority variant PDR was rarely detected in this cohort of ART-naïve pregnant women, this study points to behavioral, immunologic, pharmacologic and likely genetic factors all contributing to the likelihood that an individual's HIV replication will be quickly and sustainably suppressed by ART. Given the low prevalence of PDR with minority frequency drug resistant variants, findings from this and additional studies must be combined to further assess the risk that minority variants pose to virologic non-suppression as compared to other modifiable factors so that clinicians can optimize the care of pregnant women living with HIV to reduce mother-to-child transmission.

## Supporting information

**S1 Table. Sequences of primers used for Illumina library preparation of samples.**
(DOCX)

## Acknowledgments

The authors acknowledge the contributions of the study participants, site investigators and staff, and IMPAACT central resources who supported the P1081 study. Study drugs were provided by Merck and Company, Bristol Myers Squibb, and ViiV Healthcare.

## Author Contributions

**Data curation:** Ceejay L. Boyce, Jackson J. Wallner, R. Leavitt Morrison, David E. Shapiro.

**Formal analysis:** Ceejay L. Boyce, Ingrid A. Beck, R. Leavitt Morrison, David E. Shapiro, Lisa M. Frenkel.

**Investigation:** Sheila M. Styrchak, Samantha R. Hardy.

**Methodology:** Ceejay L. Boyce, Ingrid A. Beck, Jackson J. Wallner, Ross S. Milne, R. Leavitt Morrison, David E. Shapiro, Esaú C. João, Lisa M. Frenkel.

**Project administration:** Ceejay L. Boyce.

**Writing – original draft:** Ceejay L. Boyce, Ingrid A. Beck, Lisa M. Frenkel.

**Writing – review & editing:** Ceejay L. Boyce, Ingrid A. Beck, Sheila M. Styrchak, Samantha R. Hardy, Jackson J. Wallner, Ross S. Milne, R. Leavitt Morrison, David E. Shapiro, Esaú C. João, Mark H. Mirochnick, Lisa M. Frenkel.

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
