## [Decision Letter · Decision Letter 0]

29 Jun 2022

PONE-D-22-13194Assessment of minority frequency pretreatment HIV drug-resistant variants in pregnant women and associations with virologic non-suppression at termPLOS ONE

Dear Dr. Frenkel,

Thank you for submitting your manuscript to PLOS ONE. After careful consideration, we feel that it has merit but does not fully meet PLOS ONE’s publication criteria as it currently stands. Therefore, we invite you to submit a revised version of the manuscript that addresses the points raised during the review process.

We look forward to receiving your revised manuscript.

Kind regards,

Jason T. Blackard, PhD

Academic Editor

PLOS ONE

Journal Requirements:

Additional Editor Comments:

This is a case-control study of pre-treatment drug resistance mutations in pregnant women.  Overall, the manuscript is well written and informative.  The methods and results are well described.  Minor revisions – as well as the comments raised by the two reviewers – would strengthen the manuscript further.

The inclusion of 21 cases and 42 controls constitutes a reasonable study population size.

How many women were enrolled in the NICHD P1081?  Are they all the same HIV subtype?

Reviewers' comments:

Reviewer's Responses to Questions

**Comments to the Author**

1. Is the manuscript technically sound, and do the data support the conclusions?

Reviewer #1: Partly

Reviewer #2: Yes

2. Has the statistical analysis been performed appropriately and rigorously? 

Reviewer #1: Yes

Reviewer #2: Yes

3. Have the authors made all data underlying the findings in their manuscript fully available?

Reviewer #1: Yes

Reviewer #2: Yes

4. Is the manuscript presented in an intelligible fashion and written in standard English?

Reviewer #1: Yes

Reviewer #2: Yes

5. Review Comments to the Author

Reviewer #1: In this case-control study. there is concern that the controls are solely selected from trial participants, whereas cases may have been evaluated for trial inclusion but excluded based on primary drug resistance (PDR) by Sanger sequence. It is therefore important to disclose how many of the cases had PDR by Sanger and additional sensitivity analyses may be appropriate.

The late enrollment of women last trimester) with inadequate time for viral suppression is important and worthy of additional discussion. The time of enrollment, drug regimen and Sanger sequence data should be provided in Table 2.

The rates of PDR in the study population is compared to other studies (lines 182-184) but this comparison may not be appropriate. Most PDR studies in the literature are based on populations identified by non viral suppression whereas two-thirds of the study population were controls selected for non-viral suppression.

Reviewer #2: The paper by Boyce et al. entitled “Assessment of minority frequency pretreatment HIV drug-resistant variants in pregnant women and associations with virologic non-suppression at term” raises an important question on potential prognostic value of minority HIV drug-resistant mutations during prevention of mother-to-child transmission. Using the data generated during the IMPAACT P1081 study, the authors address whether pre-treatment minority resistance mutations in pregnant women contribute to virologic non-suppression at delivery. The study found no significant contribution of minority pre-treatment DRMs to virologic failure at delivery, which allowed the authors to conclude that testing for minority drug-resistant mutations among pregnant women might be unnecessary. The study is important, as there are little data on the topic. At the same time, small sample size, small number and heterogeneity of identified minority drug-resistant mutations accompanied by differences between cases and controls are limitations of the study that weaken the strength of major authors’ conclusion.

Critique:

- Limitations of the study need to be outlined and discussed.

- Cases differed from controls at entry by higher viral load, shorter time to delivery and predominantly EFV-based regimen. To what extent such differences could contribute to a lack of prognostic value of minority mutations?

- If time between detection of minority mutations and delivery was short, it could explain why minority mutations were not developed into major drug resistant mutations. Did the authors observe any association? Even if no, such a discussion could be useful. Providing literature regarding the time necessary for transformation of detected minority mutations into major mutations would be helpful.

- Line 159: there is a reference to Table 1. Should it be a reference to Table 2?

6. PLOS authors have the option to publish the peer review history of their article (what does this mean?). If published, this will include your full peer review and any attached files.

Reviewer #1: No

Reviewer #2: No

---

## [Author Response · Author response to Decision Letter 0]

10 Aug 2022

Editor’s Questions: “How many women were enrolled in the NICHD P1081? Are they all the same HIV subtype?”

Response: We appreciate these questions and have revised the methods section (page 4) to include more details on the parent P1081 trial. A total of 408 pregnant women (206 randomized to raltegravir, 202 to efavirenz) were enrolled at clinical sites in six different countries: Argentina (N=20), Brazil (N=190), South Africa (N=60), Tanzania (N=84), Thailand (N=47), and the US (N=7). HIV subtypes varied by sites, with subtypes B, C, A, D, and CRF01_AE known to be circulating in the six countries from which participants enrolled.

Reviewer 1’s comment #1: “In this case-control study. there is concern that the controls are solely selected from trial participants, whereas cases may have been evaluated for trial inclusion but excluded based on primary drug resistance (PDR) by Sanger sequence. It is therefore important to disclose how many of the cases had PDR by Sanger and additional sensitivity analyses may be appropriate.” 

Response: Reviewer 1’s comments highlight ambiguity in our description of participant selection in our substudy. We have revised the methods section (lines 76-77) to provide clarity on this point. All study participants, despite having detectable PDR at entry, were followed to term; however, those with PDR at entry, who were allowed to change antiretrovirals to optimize their regimen based on the PDR, and those with entry plasma HIV RNA loads <200 copies/mL were excluded from the primary analyses for the main P1081 study. Secondary analyses of the parent study included all participants regardless of PDR and switches in antiretrovirals. We selected our cases and controls from all the participants regardless of PDR at study entry and thus did not perform sensitivity analyses as suggested by this reviewer.

Reviewer 1’s comment #2: “The late enrollment of women last trimester) with inadequate time for viral suppression is important and worthy of additional discussion. The time of enrollment, drug regimen and Sanger sequence data should be provided in Table 2.”

Response: We have made modifications to Table 2 and the Discussion to include this reviewer’s suggestions. The revised Table 2 now indicates the number of days each participant with PDR was on ART prior to delivery, which study drug regimen they received, and which mutations were found by Illumina and Sanger sequencing. Any discordance between Illumina and Sanger data is described in the footnotes of the table as well as the Results section. The Discussion now includes additional verbiage (page 12) regarding late enrollment and time to suppression.

Reviewer 1’s comment #3: “The rates of PDR in the study population is compared to other studies (lines 182-184) but this comparison may not be appropriate. Most PDR studies in the literature are based on populations identified by non viral suppression whereas two-thirds of the study population were controls selected for non-viral suppression.”

Response: This reviewer’s comment raises a valid concern. However, the rates of PDR the authors use (references 12, 16-19) for comparison with this study’s rate of PDR are all from studies examining the prevalence of PDR among people initiating first-line ART without consideration of treatment outcome. Thus, the authors believe these are appropriate comparisons.

Reviewer 2’s comment #1: “Limitations of the study need to be outlined and discussed.”

Response: We thank this reviewer for their suggestion and have modified the Discussion to include a paragraph (pages 13-14) outlining and expanding on the limitations of our study.

Reviewer 2’s comment #2: “Cases differed from controls at entry by higher viral load, shorter time to delivery and predominantly EFV-based regimen. To what extent such differences could contribute to a lack of prognostic value of minority mutations?”

Response: We appreciate this reviewer’s question. While the differences between cases and controls may have limited our ability to assess the prognostic value of minority mutations by masking their effect, the rarity of minority variants in this population limited our analysis of risk posed by minority mutations and any confounding factors, making it difficult to draw any conclusions. We have noted this as a limitation of our study in the verbiage we added to the Discussion to address comment #1 (pages 13-14). 

Reviewer 2’s comment #3: “If time between detection of minority mutations and delivery was short, it could explain why minority mutations were not developed into major drug resistant mutations. Did the authors observe any association? Even if no, such a discussion could be useful. Providing literature regarding the time necessary for transformation of detected minority mutations into major mutations would be helpful.”

Response: In this study, we only performed next-generation sequencing for genotyping at study entry and thus cannot say if any minority variants present at study entry had developed into majority frequency variants by delivery. We have added this as a limitation of our study (page 14). However, because minority drug-resistant variants were only found alone in participants who achieved ART-suppression, high adherence to the ART regimen in these participants likely prevented selection pressure on these variants to become majority variants. 

Reviewer 2’s comment #4: “Line 159: there is a reference to Table 1. Should it be a reference to Table 2?”

Response: We thank this reviewer for pointing out our error. We have corrected the reference to reflect the appropriate table number.

---

## [Decision Letter · Decision Letter 1]

13 Sep 2022

Assessment of minority frequency pretreatment HIV drug-resistant variants in pregnant women and associations with virologic non-suppression at term

PONE-D-22-13194R1

Dear Dr. Frenkel,

We’re pleased to inform you that your manuscript has been judged scientifically suitable for publication and will be formally accepted for publication once it meets all outstanding technical requirements.

Kind regards,

Jason T. Blackard, PhD

Academic Editor

PLOS ONE

Additional Editor Comments (optional):

None

Reviewers' comments:

Reviewer's Responses to Questions

**Comments to the Author**

1. If the authors have adequately addressed your comments raised in a previous round of review and you feel that this manuscript is now acceptable for publication, you may indicate that here to bypass the “Comments to the Author” section, enter your conflict of interest statement in the “Confidential to Editor” section, and submit your "Accept" recommendation.

Reviewer #2: All comments have been addressed

2. Is the manuscript technically sound, and do the data support the conclusions?

Reviewer #2: Yes

3. Has the statistical analysis been performed appropriately and rigorously? 

Reviewer #2: Yes

4. Have the authors made all data underlying the findings in their manuscript fully available?

Reviewer #2: Yes

5. Is the manuscript presented in an intelligible fashion and written in standard English?

Reviewer #2: Yes

6. Review Comments to the Author

Reviewer #2: (No Response)

7. PLOS authors have the option to publish the peer review history of their article (what does this mean?). If published, this will include your full peer review and any attached files.

Reviewer #2: No

---

## [Editor Report · Acceptance letter]

16 Sep 2022

PONE-D-22-13194R1 

Assessment of minority frequency pretreatment HIV drug-resistant variants in pregnant women and associations with virologic non-suppression at term 

Dear Dr. Frenkel:

I'm pleased to inform you that your manuscript has been deemed suitable for publication in PLOS ONE. Congratulations! Your manuscript is now with our production department. 

Kind regards, 

on behalf of

Dr. Jason T. Blackard 

Academic Editor

PLOS ONE